

# Seed morphology and sculpture of invasive *Impatiens capensis* Meerb. from different habitats

Agnieszka Rewicz[1], Monika Myśliwy[2], Wojciech Adamowski[3], Marek Podlasiński[4] and Anna Bomanowska[1]

[1] Department of Geobotany and Plant Ecology, University of Lodz, Lodz, Poland
[2] Institute of Marine and Environmental Sciences, University of Szczecin, Szczecin, Poland
[3] Białowieża Geobotanical Station, Faculty of Biology, University of Warsaw, Białowieża, Poland
[4] Department of Environmental Management, West Pomeranian University of Technology, Szczecin, Poland

Corresponding author
Agnieszka Rewicz,
agnieszka.rewicz@biol.uni.lodz.pl

## ABSTRACT

*Impatiens capensis* is an annual plant native to eastern North America that is currently spreading across Europe. In Poland, due to this plant's rapid spread in the secondary range and high competitiveness in relation to native species, it is considered a locally invasive species. The microstructure of seeds is an important tool for solving various taxonomic problems and also provides data useful for determining the impact of various environmental factors on the phenotypic variability of species. This issue is particularly important in regard to invasive species which occupy a wide range of habitats in the invaded range. There are few reports on seed size and thus far no descriptions of the seed ultrastructure of *I. capensis* in the analyzed literature. We present new data on the seed morphology of *I. capensis* growing in different habitats and conditions in the secondary range of the species. The studied populations differed significantly in each of the investigated traits (seed length, width, circumference, area, roundness, and mass). Our findings showed that anthropogenic disturbances in habitats and some soil parameters (presence of carbonates, potassium, loose sand, and moisture) were statistically significant with various seed sizes and morphology in the studied populations of *I. capensis*. Moreover, our studies showed maximum seed length (5.74 mm) and width (3.21 mm) exceeding those values given in the available literature. For the first time, we also provide a detailed SEM study of the ultrastructure of the seed coat of *I. capensis*. There are two types of epidermal cells on the seeds: (a) between the ribs (elongated with straight anticlinal walls, slightly concave outer periclinal walls, and micropapillate secondary sculpture on the edges with anticyclic walls), and (b) on the ribs (isodiametric cells with straight anticlinal walls and concave outer periclinal walls). Unlike the variability of size and weight of seeds, the coat ornamentation has turned out to be a steady feature within the studied secondary range of *I. capensis*.

## INTRODUCTION

The genus *Impatiens* is the most species-rich within the family Balsaminaceae, with ca. 1,000 species distributed primarily in the Old World tropics and subtropics (*Grey-Wilson, 1980*; *Yu et al., 2015*).

    *Impatiens* has been a subject of numerous studies regarding distribution (*Zhou et al., 2019*), ecology (*Abrahamson & Hershey, 1977*; *Boyer et al., 2016*), physiology (*Nanda & Kumar, 1983*; *Tooke et al., 2005*), biochemistry (*Sreelakshmi et al., 2018*), biology (*Jacquemart et al., 2015*), pollination (*Abrahamczyk et al., 2017*), morphology (*Akiyama & Ohba, 2000*; *Janssens et al., 2018*), systematics (*Chen et al., 2007*; *Chen, Akiyama & Ohba, 2007*; *Gogoi et al., 2018*), phylogeny and evolution (*Janssens et al., 2007*; *Ruchisansakun et al., 2015*), and other (see *Adamowski, 2016–2020*). Despite the plethora of publications on various attributes of *Impatiens*, this genus requires further attention and research. *Impatiens* is taxonomically one of the most difficult groups to classify and remains a major challenge due to the enormous species richness and extraordinary morphological variability, with plants ranging from annuals growing only several centimeters high and bearing a single flower to subshrubs four meters high (*Hooker, 1904–1906*; *Grey-Wilson, 1980*; *Gogoi et al., 2018*; *Ruchisansakun et al., 2018*).

    The majority of balsam species grow in hardly accessible mountain ranges and have delicate flowers with complex morphology (*Bhaskar, 2012*; *Yu, 2012*; *Rahelivololona et al., 2018*). Herbarium specimens of balsams are difficult to prepare due to the succulent nature of the stems. Specimens need special preparations such as floral dissections (*Shui et al., 2011*) and extensive field notes, otherwise they are of limited value. Flower colors fade quickly and the position of the individual flower parts is often impossible to determine from traditionally prepared specimens. One of the taxonomically important features within the genus *Impatiens* is related to the morphology of seeds. First information on the diversity of the seed coat of *Impatiens* was reported by *Hooker & Thomson (1859)* and *Warburg & Reiche (1895)*. Other works were concerned mostly with the shape and size of seeds rather than details of their surface ornamentation (*Shimizu, 1977*).

    The development of new imaging methods enables the observation and study of ultra-small-sized structures. Scanning electron microscopy (SEM) has allowed a detailed analysis of seed coat micromorphology of *Impatiens* seeds (*Song, Yuan & Kupfer, 2005*; *Chen et al., 2007*; *Zhang et al., 2016*). Earlier works focused on seed dimensions were rarely devoted to the ultrastructure of seeds (*Shimizu, 1979*; *Lu & Chen, 1991*). The sculpture on seed coats offers a set of characters which can be used to identify a species, and in combination with other morphological data, can provide crucial evidence towards the taxonomy of a genus (*Lu & Chen, 1991*; *Song, Yuan & Kupfer, 2005*; *Utami & Shimizu, 2005*; *Cai et al., 2013*; *Yu et al., 2015*).

    Seed morphological features of *Impatiens* have not only been used for solving various taxonomic problems within the genus but also prove to be useful for determining the impact of various environmental factors on the phenotypic variability of balsam species (*Argyres & Schmitt, 1991*; *Schmitt, 1993*; *Maciejewska-Rutkowska & Janczak, 2016*). The understanding of environmentally induced variation in an individual plant phenotype

is crucial for predicting population responses to environmental changes. This issue is particularly important regarding invasive species which occupy a wide range of habitats in the invaded range (*Richards et al., 2006*).

Despite an increasing number of publications on the surface of *Impatiens* seeds by SEM (e.g., *Shimizu, 1979*; *Yu, Chen & Qin, 2007*; *Shui et al., 2011*; *Xia et al., 2019* a.o.), there is still a lack of information on the seed micromorphology of the majority of species. In fact, a detailed understanding of the seed morphology of the entire genus *Impatiens* is missing, despite major studies using novel imaging methods (e.g., *Yuan et al., 2004*; *Ruchisansakun et al., 2015*; *Rahelivololona et al., 2018*). As yet, only about 170 species have been investigated, which is about one fifth of all known balsams (*Maciejewska-Rutkowska & Janczak, 2016*).

One of the species with morphologically undescribed seeds is *Impatiens capensis* (jewelweed, orange balsam), an annual plant native to eastern North America (*Meusel et al., 1978*), which is currently spreading across Europe. Today *I. capensis* is considered as naturalized in several European countries (*Matthews et al., 2015*), including Poland, where the species is locally established and invasive due to its rapid spread in the secondary range and high competitiveness in relation to native species, even perennials (*Tokarska-Guzik et al., 2012*). In Poland, it was found for the first time in 1987 (*Pawlaczyk & Adamowski, 1991*), and it is currently spreading in the Western Pomerania region (*Popiela et al., 2015*; M Myśliwy, pers. obs., 2017). The species occurs in the area of the Szczecin Lagoon and enters alder forests, willow shrubs, rushes and riparian tall herb fringe communities (*Pawlaczyk & Adamowski, 1991*; *Myśliwy, Ciaciura & Hryniewicz, 2009*; M Myśliwy, pers. obs., 2014). It also appears in moist anthropogenic habitats, e.g., along roadside ditches (M Myśliwy, pers. obs., 2017).

*Impatiens capensis* is an annual plant growing from 0.5–1.5 m or more in height. The flowers are 2.5–3.0 cm long and orange with darker patches in the most common f. *capensis*. The lower sepal forms a light-orange nectar spur, 5–9 mm long, which is bent at 180° to lie parallel to the sepal-sac (*Zika, 2006*). Besides color, it differs from the predominantly Eurasiatic *I. noli-tangere* in that the lower sepal is more rapidly constricted into the spur and the position of the spur (*Zika, 2009*). The fruit is a five-valved capsule, 2.0–2.5 cm long and 0.3–0.5 cm wide, with explosive dehiscence ejecting the seeds (*Moore, 1968*; *Gleason & Cronquist, 1991*; *Day, Pellicer & Kynast, 2012*). The seeds are laterally compressed, prolate spheroid, with four strong ribs of 5–5.6 × 2.7–3.1 mm (*Bojňanský & Fargašová, 2007*). The weight ranges from 6.4 to 26.9 mg (*Simpson, Leck & Parker, 1985*). *Schemske (1978)* recorded 11.5 mg for cleistogamous seeds and 13.3 mg for chasmogamous ones, and *Waller (1982)* 10.6 mg. The seed surface is wrinkled or rough, lusterless, dark-brown, with some roundish and paler spots (*Bojňanský & Fargašová, 2007*).

Numerous studies (several hundred; see *Adamowski, 2016–2020* and the literature cited therein) have been devoted to the ecology, biology, and genetics of this species (e.g., *Antlfinger, 1989*; *Schmitt, Ehrhardt & Swartz, 1985*; *Donohue & Schmitt, 1999*; *Donohue et al., 2000*; *Zika, 2009*; *Tabak & Von Wettberg, 2008*; *Day, Pellicer & Kynast, 2012*). However, a review of the available literature showed a scarcity of data on seed size and a complete lack of information describing the morphological variation of the seed

**Table 1  List of the studied populations of *Impatiens capensis* Meerb. in Poland.**

| Code | Locality | Latitude | Longitude | Habitat | Average plant height [cm] | Number of analyzed seeds | Population size (mature individuals) |
|------|----------|----------|-----------|---------|---------------------------|--------------------------|--------------------------------------|
| A | Podgrodzie | 53.740222° | 14.306667° | tall herbs on the bank of Szczecin Lagoon | 130 | 29 | 20–30 |
| B | Lubin | 53.865056° | 14.426778° | tall herbs and grasses near water seeps | 50 | 24 | >20 |
| C | Unin | 53.894806° | 14.634444° | tall herbs along the river | 120 | 27 | 20–30 |
| D | Czarnocin | 53.722306° | 14.549167° | tall herbs on the bank of Szczecin Lagoon | 130 | 30 | >50 |
| E | Święta | 53.559861° | 14.659083° | tall herbs along roadside ditch | 165 | 27 | >100 |
| F | Szczecin-Zdroje | 53.382861° | 14.614944° | tall herbs along the river | 120 | 29 | >50 |
| G | Police | 53.573194° | 14.581472° | willow forest along artificial canal | 145 | 28 | >100+ |
| H | Trzebieradz | 53.675417° | 14.441444° | alder carr | 70 | 30 | >100+ |

coat of *I. capensis* (*Schemske, 1978*; *Waller, 1982*; *Simpson, Leck & Parker, 1985*; *Bojňanský & Fargašová, 2007*).

The aim of our work has been to characterize the micromorphological traits and ultrastructure of *I. capensis* seeds from various habitats and growing conditions and their morphological variability. Anthropogenic changes in habitats were expected as important factors affecting seed micromorphology and ultrastructure.

## MATERIALS AND METHODS

### Study sites

Seeds were collected from August to September 2018 (to avoid seasonal variability) from eight populations of *I. capensis* in Poland. We sampled the entire Polish range of this species from all types of habitats, from natural (alder carrs, hydrophilous tall herb communities along rivers, near water seepages, and along the banks of the Szczecin Lagoon) to anthropogenic (tall herb communities along roadside ditches, transformed forests along artificial canals) (Table 1, Fig. 1). The studied populations were also subject to different lighting conditions, which were scored using a 3-point scale: plants which grew in willow forests and the understory of alder carrs were strongly shaded (3), while those from tall herb communities were partly shaded by solitary trees (2) or exposed to full sun (1). As the height of *I. capensis* specimens, the location of capsules within the plant (main stem vs. branches), and their derivation from flowers of various types (cleistogamous vs. chasmogamous) may affect seeds weight (*Waller, 1982*), the seeds for our study were collected always from the main stems of 8–10 plants of similar (average for the population) height and from capsules derived from chasmogamous flowers, to minimize the bias. Species nomenclature was adopted from Euro+Med PlantBase (*Euro+Med PlantBase, 2019*).

### Biometric and SEM analysis

From 24 to 30 mature seeds were used from each population for biometric analysis. We measured four quantified seed traits: seed length (SL), seed width (SW), seed circumference (SC), and seed area (SA). The seeds were measured as previously described in *Rewicz et al. (2017)*. In order to describe the seed mass, we used 15 seeds from each population. The
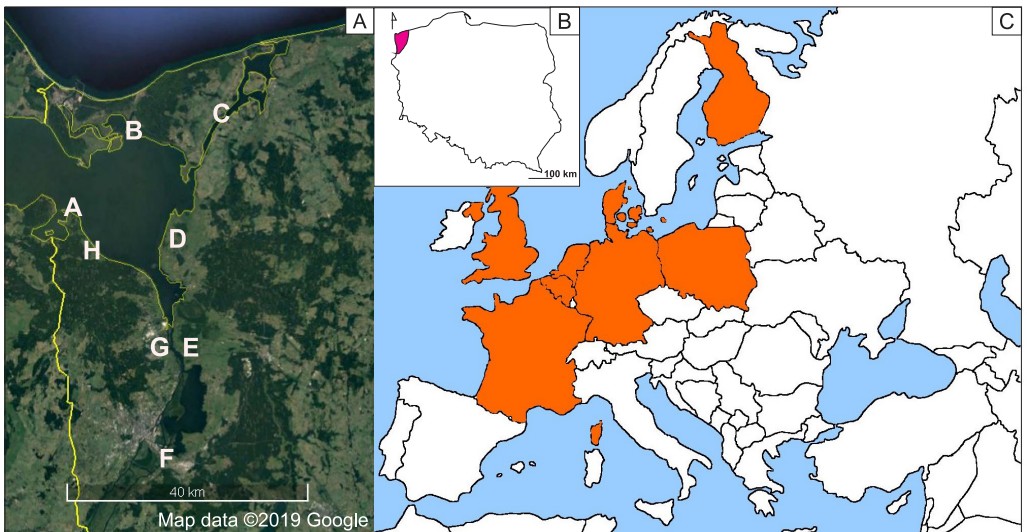

**Figure 1** **Distribution map of *Impatiens capensis* Meerb. in Europe (C), range in Poland (B), sites of the studied populations (A) (prepared by Adamowski & Myśliwy).** Satellite map data ©2019 Google, Modifed using CorelDRAW 18. For explanation of symbols, see Table 1.

seeds were weighed with an Ohaus PA 21. We determined *roundness* by the formula: R = $4 \times$ area/$\pi$ [Major axis]$^2$ as defined by *Ferreira & Wayne (2010)*.

We used eight seeds from each population for SEM. The seeds were air-dried and sputter-coated with a 4-nm-thick layer of gold (Leica EM ACE200). The SEM work was performed on a Phenom Pro X Scanning Electron Microscope at the Department of Invertebrate Zoology and Hydrobiology, University of Lodz, Poland. The 3D models of the seed surface were generated using the dedicated software *3D Roughness Reconstruction* for Phenom. SEM micrographs were analyzed as previously described in *Rewicz et al. (2017)*. Seed shape terminology and types of seed surfaces were adopted from *Barthlott (1981)*.

## Soil properties

In order to characterize habitat conditions at each locality, five soil core samples (0–20 cm depth) were collected and then mixed together into one composite sample. Soil samples were dried at room temperature, and passed through a sieve to remove fractions larger than one mm. The following physicochemical soil parameters were determined (*Bednarek et al., 2011*), as first described by *Myśliwy (2019)*: organic matter content defined as the loss on ignition (LOI)–soil samples annealed at 550 °C (%); grain composition (the content of sand, silt, clay)–Bouyoucos's sedimentation method, modified by Casagrande and Prószyński; granulometric categories according to the *Polish Society of Soil Science (2009)* classification; soil reaction (pH)–the potentiometric method, in 1-M solution of KCl; soil calcium carbonate ($CaCO_3$) content (%)–the Scheibler's method; organic carbon ($C_{org}$) content (%), and total nitrogen ($N_{tot}$) content (%) were determined by an elemental analyzer CHNS/O FlashSmart (Thermo Scientific), and the C/N ratio; the content of available forms of soil nutrients (mg/100 g soil): calcium (Ca) and sodium (Na)

were determined spectrophotometrically (Ca–AAS and Na–EAS) on ICE3000; potassium (K) and phosphorus (P)–measured according to the Egner-Riehm method; magnesium (Mg)–measured by Schachtschabel's method; soil moisture content, hand-felt assessed directly in the field using a 4-point scale recommended by the *Soil Science Society of Poland (2017)*: (1) dry (the soil crumbles and turns to dust, it is neither cool nor moist to touch; it darkens visibly after wetting), (2) fresh (the soil feels cool, but no moisture is felt; darkens after wetting), (3) moist (the soil moistens fingers and tissue paper, but water does not leak when squeezed; clayey, loamy, and some dusty soils are plastic; does not darken after wetting), (4) wet (water leaks from the soil when squeezed, aggregates, soil smears).

## Data analysis

The five following basic characteristic traits were calculated: arithmetic average (x), minimum and maximum values (min/max), coefficient of variation (CV), and standard deviation (SD). The distribution of the data was not normal; statistical analysis was based on the Kruskal-Wallis test (for $p \leq 0.05$), which is a nonparametric alternative to ANOVA (*Zar, 1984*). Correlation between pairs of morphological characters was evaluated using Spearman's correlation coefficient and the values were adopted after *Meissner (2010)*, (correlation: less than 0.20–very poor; 0.21–0.39–weak; 0.40–0.69–moderate; 0.70–0.89– strong; and above 0.89–very strong).

The cluster analysis based on the nearest neighbor method was performed using the matrix on the population's mean values. As the dataset required a linear response model (*Jongman et al., 1995*), the Redundancy Analysis (RDA) was used to relate the variability of morphological traits of seeds to environmental variables. The variables $C_{org}$ and $N_{tot}$ were excluded from the RDA as they were strongly correlated with organic matter content (LOI). The Monte Carlo permutation test with the forward selection of environmental variables was applied to determine the importance and statistical significance of variables in explaining the variability in seeds. The software packages Canoco v.4.5 (*Ter Braak & Šmilauer, 2002*), MVSP 3.2 (*Kovach, 2010*), and STATISTICA PL. ver. 13.1 (*StatSoft Inc, 2011*) were used for all analyses (*Van Emden, 2008*; *Lepš & Šmilauer, 2003*).

# RESULTS

## Biometric analysis

Seeds from the G (Police) population were the largest, with average values of length (SL) 4.60 mm, width (SW) 2.71 mm, circumference (SC) 11.65 mm, and area (SA) 9.26 mm$^2$; comparatively large seeds were also obtained from the E population (Święta); the B (Lubin) population had the shortest (mean SL 3.88 mm) and narrowest seeds (mean SW 2.03 mm) (Table 2).

The minimum values of analyzed traits were also recorded in the B (Lubin) population (SL 3.16 mm, SW 1.12 mm, SC 7.27 mm, SA 2.43 mm$^2$). The maximum values of length (5.74 mm), circumference (14.59 mm), and area (13.54 mm$^2$) were recorded in the G population (Police).

A very strong Spearman correlation ($r = 0.94$) was observed between the seed area and circumference (Table 3). The most variable features were the seed area (CV = 21.76%)

**Table 2  Biometric comparison of seed traits of *Impatiens capensis* Meerb.** Seed length (SL), seed width (SW), seed circumference (SC), seed area (SA), variation coefficient (CV), standard deviation (SD), minimum/maximum (Min/Max), arithmetic average (X), A–H as in Table 1.

| | A | B | C | D | E | F | G | H | x |
|---|---|---|---|---|---|---|---|---|---|
| Weight (mg) | 7.66 | 6.52 | 8.16 | 7.82 | 9.82 | 8.62 | **11.42** | 6.92 | 8.37 |
| SL (mm) | 4.05 | 3.88 | 4.23 | 4.11 | 4.46 | 4.17 | **4.60** | 4.41 | 4.24 |
| Min-max | 3.50-4.64 | 3.16-4.48 | 3.68-4.70 | 3.59-4.75 | 3.85-5.26 | 3.43-4.73 | 3.88-5.74 | 3.59-4.82 | 3.16-5.74 |
| SD | 0.26 | 0.40 | 0.29 | 0.30 | 0.32 | 0.40 | 0.40 | 0.27 | 0.40 |
| CV | 6.50 | 10.28 | 6.95 | 7.35 | 7.21 | 9.71 | 8.75 | 6.19 | 9.34 |
| SW (mm) | 2.23 | 2.03 | 2.36 | 2.56 | 2.60 | 2.40 | **2.71** | 2.23 | 2.39 |
| Min-max | 1.53-2.82 | 1.12-2.61 | 1.78-3.00 | 2.14-2.94 | 2.19-3.33 | 1.88-2.99 | 2.15-3.21 | 1.71-2.93 | 1.12-3.33 |
| SD | 0.28 | 0.42 | 0.33 | 0.19 | 0.31 | 0.32 | 0.30 | 0.30 | 0.37 |
| CV | 12.50 | 20.57 | 13.98 | 7.30 | 12.02 | 13.19 | 11.04 | 13.51 | **15.35** |
| SC (mm) | 10.00 | 9.55 | 10.51 | 10.61 | 11.21 | 10.49 | **11.65** | 10.75 | 10.61 |
| Min-max | 8.63-11.94 | 7.27-10.98 | 9.13-11.70 | 9.35-12.70 | 9.77-13.27 | 8.61-12.10 | 10.17-14.59 | 8.9-12.20 | 7.27-14.59 |
| SD | 0.77 | 1.09 | 0.67 | 0.69 | 0.89 | 0.89 | 1.04 | 0.67 | 1.02 |
| CV | 7.65 | 11.42 | 6.39 | 6.50 | 7.90 | 8.53 | 8.97 | 6.22 | 9.64 |
| SA (mm$^2$) | 6.52 | 5.79 | 7.15 | 7.71 | 8.42 | 6.74 | **9.26** | 7.25 | 7.44 |
| Min-max | 4.72-9.53 | 2.43-7.69 | 5.10-9.10 | 5.83-1.23 | 6.15-12.16 | 4.82-1.21 | 6.74-13.54 | 5.27-9.61 | 2.43-13.54 |
| SD | 1.14 | 1.51 | 1.21 | 1.07 | 1.48 | 1.27 | 1.63 | 1.08 | 1.62 |
| CV | 17.54 | 26.06 | 16.92 | 13.88 | 17.63 | 17.58 | 17.65 | 14.96 | **21.76** |

**Table 3  Spearman correlation values for seed traits of *Impatiens capensis* Meerb.** All values with significance of $p < 0.05$.

| | Length | Width | Circumference | Area |
|---|---|---|---|---|
| Length | 1.00 | 0.47 | 0.85 | 0.72 |
| Width | | 1.00 | 0.73 | 0.84 |
| Circumference | | | 1.00 | 0.94 |
| Area | | | | 1.00 |

and width (CV = 15.35%). The variation of seed traits ranged insignificantly from 6.19% (H population) to 10.28% (B) for SL; from 7.30% (D) to 20.57% (B) for SW; from 6.22% (H) to 11.42% (B) for SC; and from 13.88% (D) to 26.06% (B) for SA, respectively.

The G (Police: 11.42 mg) and E (Święta: 9.82 mg) populations are characterized by the heaviest seeds. The lightest seeds were observed in the following populations: B (Lubin: 6.52 mg) and H (Trzebieradz: 6.92 mg) (Fig. 2, Table 2).

The Kruskal-Wallis test showed that the *I. capensis* populations differed significantly in each of the analyzed traits. The conducted *post hoc* test (DunnTest) showed that the populations from: Police (G), followed by Czarnocin (D), Święta (E), and Trzebieradz (H) showed the greatest variation in terms of studied traits among all the populations (Table 4).

The similarity analysis using Euclidean's distances showed two main clusters (Fig. 3). The first cluster included six populations of *I. capensis* (A–D, F, H), all derived from natural habitats, while the other cluster groups two populations (E, G) from anthropogenic habitats, where the examined plants were the highest (Table 1). According to the dendrogram (Fig. 3), populations C and F are the closest to each other; both were associated with river valleys

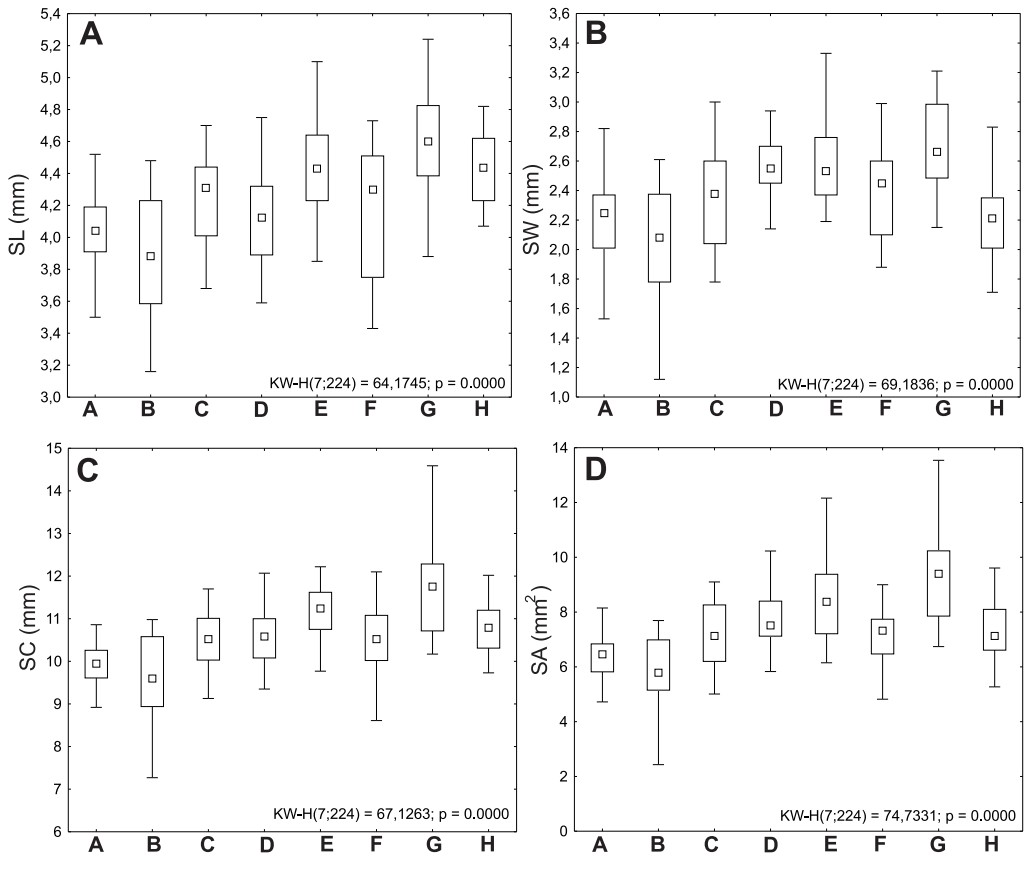

**Figure 2   Ranges of variation of seed traits of *Impatiens capensis* Meerb.** The boxes represent the 25th–75th percentiles; the upper and lower whiskers extend the minimum and maximum data point; the square inside the box indicates median. (A) Seed length (SL); (B) seed width (SW); (C) seed circumference (SC); (D) seed area (SA).

**Table 4   Interpopulation variability for length (A), width (B), circumference (C), and area (D) of seeds of *Impatiens capensis*.** Asterisk next to letter indicates significance at $p < 0.05$.

|                | Podgrodzie | Lubin | Unin    | Czarnocin | Święta     | Szczecin-Zdroje | Police     | Trzebieradz |
|----------------|------------|-------|---------|-----------|------------|-----------------|------------|-------------|
| Podgrodzie     |            | ABCD  | ABCD    | AB*CD*    | A*B*C*D*   | ABCD            | A*B*C*D*   | A*BC*D      |
| Lubin          |            |       | ABCD    | AB*C*D*   | A*B*C*D*   | AB*CD           | A*B*C*D*   | A*BC*D      |
| Unin           |            |       |         | ABCD      | ABCD       | ABCD            | A*B*C*D*   | ABCD        |
| Czarnocin      |            |       |         |           | A*BCD      | ABCD            | A*BC*D     | A*B*CD      |
| Święta         |            |       |         |           |            | ABCD            | ABCD       | AB*CD       |
| Szczecin-Zdroje |           |       |         |           |            |                 | A*BC*D*    | ABCD        |
| Police         |            |       |         |           |            |                 |            | AB*CD*      |
| Trzebieradz    |            |       |         |           |            |                 |            |             |

(Dziwna and Oder rivers, respectively) and close to the river bed, hence under the influence of flooding. The D and A populations were growing in tall herb communities on the banks of the Szczecin Lagoon. The most distinct populations in the first cluster (H and B) were

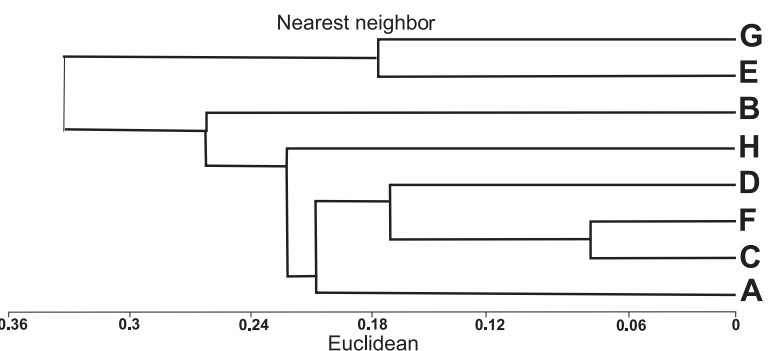

**Figure 3** Dendrogram of similarities of populations of *Impatiens capensis* Meerb. in Poland, obtained by the nearest neighbor method.

**Table 5** Seed roundness (R) comparison between populations of *Impatiens capensis* Meerb. in Poland.

|   | R | SD | A | B | C | D | E | F | G | H |
|---|---|----|---|---|---|---|---|---|---|---|
| A | 0.50 | 0.06 | a | a | a | $p = 0.0023$ | a | a | a | a |
| B | 0.48 | 0.08 |   | a | a | $p = 0.0000$ | a |   | $p = 0.0087$ | a |
| C | 0.51 | 0.09 |   |   |   | $p = 0.0033$ | a | a | a | a |
| D | 0.58 | 0.06 |   |   |   |   | a | a | a | $p = 0.0000$ |
| E | 0.54 | 0.07 |   |   |   |   |   | a | a | $p = 0.0197$ |
| F | 0.53 | 0.07 |   |   |   |   |   |   | a | a |
| G | 0.56 | 0.07 |   |   |   |   |   |   |   | $p = 0.0002$ |
| H | 0.47 | 0.05 |   |   |   |   |   |   |   |   |

**Notes.**
SD, standard deviation, A–H, in Table 1; the same capital letters mean the values do not differ significantly.

also found on the banks of the Szczecin Lagoon, but they had the lowest average height and differed in habitat conditions from the other populations of this cluster (Table 1).

The investigated morphological parameter of seed shape, roundness, showed statistically significant differences between the populations ($p < 0.05$). For roundness, the highest value was recorded at D: Czarnocin (0.58) (tall herbs on the bank of the Szczecin Lagoon) and the lowest at H: Trzebieradz (0.47) (alder carr) (Table 5).

## Biometric variability of seeds and its relationship with environmental conditions

All environmental variables included in the RDA accounted for 35.6% of the total variation. The results of stepwise forward selection of variables indicated that five variables: anthropogenic disturbances (Anthrop), carbonates ($CaCO_3$), loose sand presence (LoSa), potassium (K), and soil moisture content (Moist) were statistically significant and varied between the studied populations of *I. capensis* (Fig. 4). Along the gradient represented by Axis I, the highest correlation between the sample position and environmental variables (the so-called interset correlation) was typical of anthropogenic disturbances and $CaCO_3$,

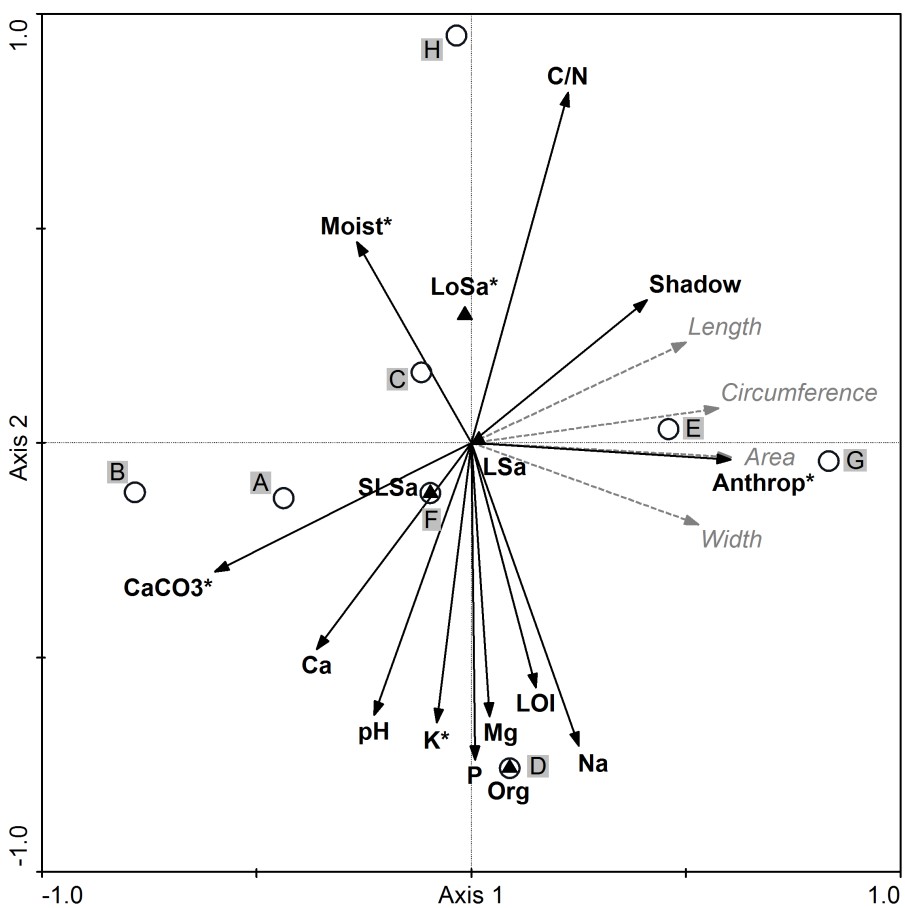

**Figure 4  Ordination diagram of populations of *Impatiens capensis* Meerb.** (A–H) Seed biometric traits (dotted gray arrows) and environmental variables (solid black arrows) along the first two RDA axes. * Statistically significant variables; Anthrop, anthropogenic disturbances; shadow, degree of shading. For codes of populations and soil properties see Tables 1 and 6, respectively.

followed by the degree of shading and soil Ca, while the C/N ratio was most closely correlated with Axis II, followed by soil content of P, Na, K, and organic soil.

The location of population H (Trzebieradz) in the ordination space (the upper part of the RDA diagram) was associated with the highest C/N ratio, the highest soil moisture and shading, as well as with the lowest soil pH and the lowest soil content of $CaCO_3$, Ca, P, and K (Fig. 4, Table 6). At the same time, the H population was dominated by short specimens (Table 1), with one of the lightest seeds and average values of biometric traits (Table 2). In contrast, populations characterized by the longest, widest, and heaviest seeds, from E (Święta) and G (Police), located in the right-hand side of the RDA diagram, were also related to a relatively high C/N ratio, but unlike the previous population, they were associated with a low level of soil moisture as well as the highest anthropogenic disturbances (Fig. 4), and consisted of the tallest specimens (Table 1). Population D (Czarnocin) occupied the bottom part of the diagram and was distinct in its organic soil, with the highest content of organic matter (LOI), as well as P, K, Mg, and Na content in the soil, while having the

**Table 6** Soil conditions at the investigated sites of occurrence of *Impatiens capensis* Meerb. in Poland.

| Code/Locality | Soil group | LOI | $C_{org}$ | $N_{tot}$ | C/N | pH | $CaCO_3$ | Ca | P | K | Mg | Na | Moist |
|---|---|---|---|---|---|---|---|---|---|---|---|---|---|
| A | LoSa | 11.6 | 7.11 | 0.52 | 13.61 | 6.6 | 0.00 | 3007 | 60.2 | 68.3 | 378.0 | 405.4 | 3 |
| B | LSa | 15.8 | 9.98 | 0.65 | 15.37 | 7.4 | 3.54 | 30769 | 194.5 | 171.8 | 258.0 | 181.6 | 2 |
| C | LSa | 23.4 | 12.63 | 0.95 | 13.25 | 7.3 | 2.31 | 18476 | 130.4 | 206.4 | 325.0 | 177.9 | 3 |
| D | Org | 65.3 | 31.90 | 2.54 | 12.54 | 6.7 | 1.47 | 21816 | 653.6 | 310.0 | 1796.0 | 1152.8 | 3 |
| E | LoSa | 19.5 | 13.18 | 0.81 | 16.25 | 6.3 | 0.00 | 9142 | 67.6 | 187.9 | 204.0 | 62.2 | 2 |
| F | SLSa | 6.0 | 3.70 | 0.27 | 13.49 | 7.5 | 1.09 | 5792 | 111.6 | 85.9 | 89.0 | 44.0 | 3 |
| G | LSa | 17.3 | 8.77 | 0.55 | 15.84 | 6.8 | 0.00 | 7801 | 117.7 | 47.4 | 264.0 | 638.5 | 2 |
| H | LoSa | 16.5 | 7.95 | 0.43 | 18.69 | 5.1 | 0.00 | 2705 | 31.4 | 33.7 | 262.0 | 57.6 | 4 |

Notes.

LoSa, loose sand; LSa, loamy sand; SLSa, slightly loamy sand; Org, organic soil; LOI, organic matter content; $C_{org}$, organic carbon; $N_{tot}$, total nitrogen; C/N, C/N ratio; pH, soil pH; $CaCO_3$, carbonates; Ca, calcium; P, phosphorus; K, potassium; Mg, magnesium; Na, sodium; Moist, soil moisture content.

lowest C/N ratio (Table 6). The lowest values of the seed biometric traits were found for population B (Lubin), located in the left-hand part of the RDA diagram (Fig. 4, Table 2), and associated with high soil pH and the highest content of soil carbonates and calcium, as well as a low level of soil moisture (Table 6). The other populations (A: Podgrodzie; C: Unin; F: Szczecin-Zdroje) were also on the left side of the diagram, but closer to the center (Fig. 4). Neither their seed biometric traits nor habitat conditions were distinct (Tables 2, 6).

### Seed surface ultrastructure

The studied seeds of *I. capensis* were round in shape, with a lusterless, rough, and dark-brown surface, without roundish and paler spots (Fig. 5). The seeds had four strong, clear ribs, the apex and bottom narrowed. Each rib was built of rows of 4–5 cells and had a darker color than the surface between them (Fig. 5H). The seed coat is composed of two types of epidermal cells (Figs. 5E, 5H) creating a net-like pattern. The cells of the seed surface between the ribs were: elongated with straight anticlinal walls (Fig. 5E), raised cell boundaries between the cells (Fig. 5G), slightly concave outer periclinal walls (Figs. 5F, 5G) and a micropapillate secondary sculpture on the edges of anticyclic walls (Fig. 5F). Near the ribs, there were rows of 4–7 isodiametric cells (Figs. 5I, 5K) with straight anticlinal walls (Figs. 5L), with raised cell boundaries (Fig. 5M) and concave outer periclinal walls. Seeds from all studied populations did not differ in their ultrastructure (Figs. 5A–5D).

### DISCUSSION

SEM gives us the means for studying the morphological characters of seeds and their ultrastructural characteristics which helps or identifying and determining the taxonomic delimitation of various angiosperm groups, as demonstrated for Brasssicaceae (*Tantaway et al., 2004*), Caryophyllaceae (*Ullah et al., 2019a*; *Ullah et al., 2019b*), Poaceae (*Martín-Gómez et al., 2019*), Cyperaceae (*Więcław et al., 2017*), Ranunculaceae (*Constantinidis, Psaras & Kamari, 2001*; *Rewicz et al., 2017*; *Martín-Gómez, Rewicz & CerVantes, 2019*; *Hadidchi, Attar & Ullah, 2020*), Rosaceae (*Ballian & Mujagić-Pašić, 2013*), Cervantes

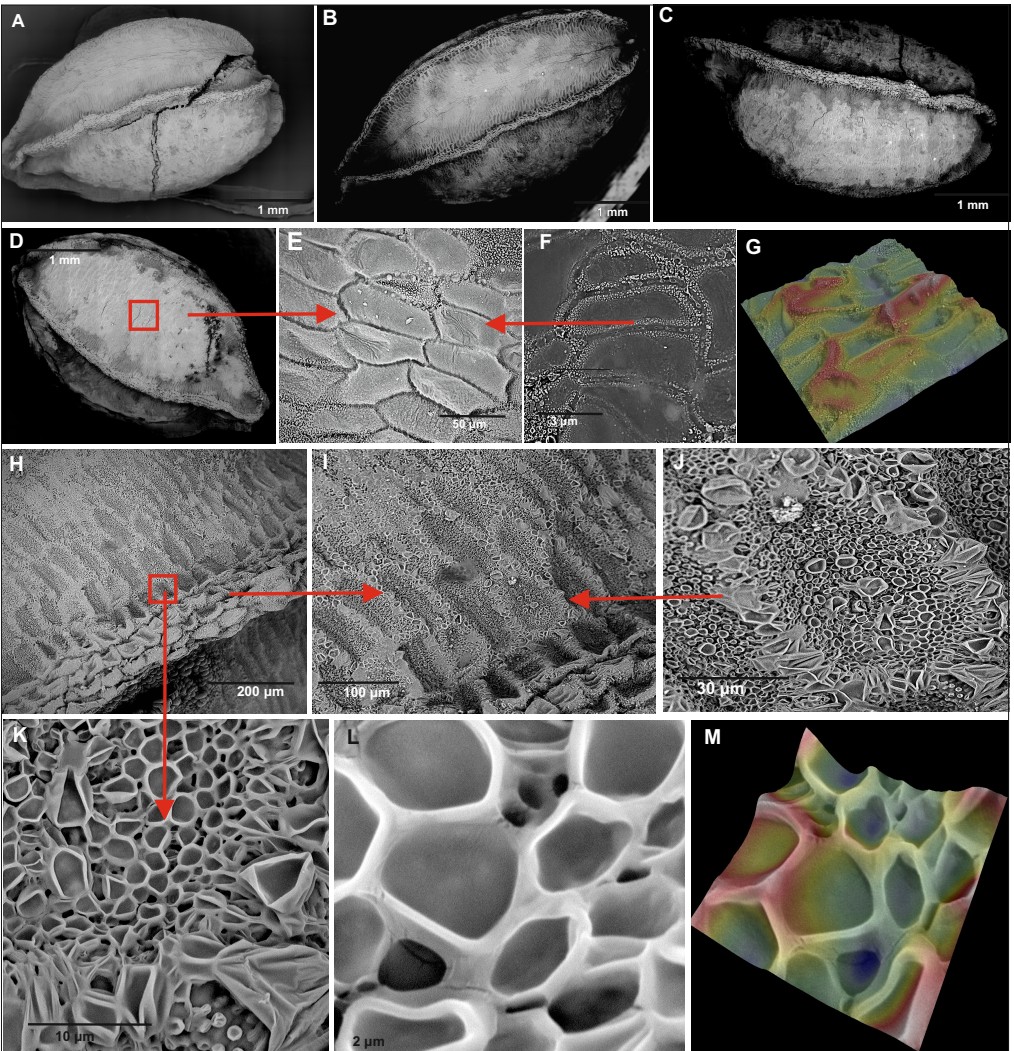

**Figure 5** Comparison of seeds of *Impatiens capensis* Meerb. (A–D) General view of seeds from (A) Czarnocin, (B) Police, (C) Lublin, (D) Podgrodzie; (E–F) seed sculpture; (G) seed surface 3D ultrastructure between the ribs; (H) rib; (I–L) cells near rib; (M) 3D ultrastructure of cells near rib.

(*Akbari & Azizian, 2006*), and Orchidaceae (*Gamarra et al. 2007*; *Gamarra et al., 2010*; *Rewicz, Kołodziejek & Jakubska-Busse, 2016*). Although seed morphology alone does not provide universally applicable key characters for species identification, it can be as helpful as many other characters used in taxonomy.

Members of Balsaminaceae have a diverse and elaborately sculptured seed coat. Unfortunately, till now seed morphology has been observed only for a small number of *Impatiens* species, which has limited the use of the morphological traits of seeds in taxonomy and classification (e.g., *Song, Yuan & Kupfer, 2005*; *Utami & Shimizu, 2005*; *Chen et al., 2007*; *Yu, Chen & Qin, 2007*; *Jin et al., 2008*; *Shui et al., 2011*).

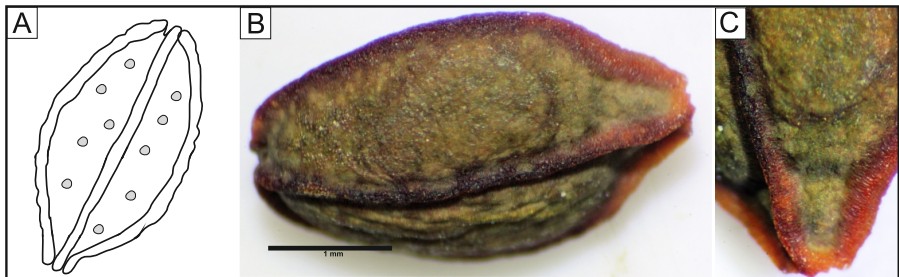

**Figure 6** (A) Drawing of seed of *Impatiens capensis*. (based on *Bojňanský & Fargašová, 2007*), seed of *Impatiens capensis* under the light microscope; (B) general view; (C) rib.

We here provide new data about the seed morphology and seed coat sculpture of *I. capensis,* as well as new information about the area of its distribution in Poland. Our studies have shown maximum seed length (5.74 mm) and width (3.21 mm) beyond the values reported elsewhere. *Bojňanský & Fargašová (2007)* found seeds of *I. capensis* to be 5-5.6 mm long and 2.7–3.1 mm wide. Our ultrastructural studies have shown two types of cells on between the ribs and on the ribs, that have previously not been described (Fig. 5). The occurrence of several types of epidermal cells on the seeds of members of *Impatiens* was previously noted, for instance, three types of epidermal cells have been reported in *Impatiens aconitoides* by *Shui et al. (2011)*. We were not able to confirm in any of our studied populations the presence of roundish spots on the surface of seeds (Fig. 6) as reported by *Bojňanský & Fargašová (2007)*, which may be due to the different geographical origin of the examined seeds: our seeds of *I. capensis* are from wild-growing populations from various habitats, while those studied by *Bojňanský & Fargašová (2007)* were obtained from cultivation and of unknown origin.

The analysis of SEM micrographs of *I. noli-tangere* seeds closely related to *I. capensis* (*Yu et al., 2015*) has shown that seed coats of this species vary significantly depending on the geographical origin of the seeds (*Utami & Shimizu, 2005*; *Chen et al., 2007a*; *Jin et al., 2008*). On the other hand, the comparison of the seed micromorphology of *I. capensis* has not shown similarity to seed coat ornamentation of the aforementioned *I. noli-tangere* (with narrow and ellipsoid seeds, fine reticulate subtype, testa cells with reticulate thickened outer walls; *Song, Yuan & Kupfer, 2005*; *Utami & Shimizu, 2005*; *Chen et al., 2007*; *Jin et al., 2008*). Despite the fact that both species are closely related and may be confused (*Zika, 2009*; *Yu et al., 2015*), their seeds clearly differ morphologically. The new data presented here may be useful in the identification of these species. In turn, there is no information about the seed morphology of *I. pallida*, which is sympatric and synchronic species to *I. capensis* (*Rust, 1977*), which makes this subject even more difficult. Elucidating the overall variation in seed coat micromorphology and to implement this feature to taxonomy of *I. capensis* will require more samplings, also within the native range of orange jewelweed as well as other closely related species and this eventually should become the basis for further comparisons and studies. Seed ultrastructure appears to be a constant feature within a taxonomic unit (*Stace, 1992*) and, as morphological studies show, seed shape and size are highly diverse at

the genus and species levels (*Yu, Chen & Qin, 2007*; *Jin et al., 2008*; *Shui et al., 2011*; *Ullah et al., 2019a*; *Ullah et al., 2019b*; *Hadidchi, Attar & Ullah, 2020*). Both statements have been proven for *I. capensis* in Poland.

Data concerning the size, shape and structure of seeds not only have been used as an important tool for solving various taxonomic problems within the genus *Impatiens* but also provide results useful for determining the impact of various environmental factors on the phenotypic variability of these species (*Bell, Lechowicz & Schoen, 1991*; *Argyres & Schmitt, 1991*; *Schmitt, 1993*; *Chmura, Csontos & Sendek, 2013*; *Maciejewska-Rutkowska & Janczak, 2016*).

Environmental heterogeneity is indicated as a major factor driving morphological changes (*Nakazato, Bogonovich & Moyle, 2008*). Seeds are sensitive to changes in biotic and abiotic conditions (*Moles et al., 2005*). According to *Silvertown (1989)*, the correlation between seed size and the place where plant is growing is an adaptive feature. Bigger seeds occur in habitats with stable environmental conditions, where seedlings may grow slowly. Small seeds are generally produced by plants with a short life cycle, growing mainly in disturbed habitats.

Orange balsam is known for colonizing a wide range of habitats (*Schemske, 1978*; *Waller, 1980*). Moreover, *Simpson, Leck & Parker (1985)* have shown that *I. capensis* vegetative and reproductive growth parameters reflect habitat differences. Light availability (*Simpson, Leck & Parker, 1985*) as well as soil moisture and pH (*Waller, 1980*) have been reported to affect its growth patterns. Our studies indicate that five environmental variables were statistically significant and were able to serve to discern the studied populations in terms of seed size and weight: anthropogenic disturbances (which may serve as a proxy for habitat fertility), carbonates ($CaCO_3$), loose sand presence, potassium (K), and soil moisture (Fig. 4). Populations G (Police) and E (Święta), occurring in the most disturbed anthropogenic habitats (artificial canal and roadside), have the heaviest seeds as a result of growth under favorable environmental conditions (neutral or slightly acidic soil with a relatively high C/N ratio). In turn, population B (Lubin) with the smallest and lightest seeds was associated with high soil pH, and the highest content of soil carbonates and calcium. Interestingly, *Waller (1982)* reported that the higher nodes of *I. capensis* individuals tended to produce heavier seeds. In *Waller's (1982)* opinion, the position effect probably leads to a greater mean seed size for higher plants. *Werner & Platt (1976)* stated that populations growing at higher plant densities often produce larger seeds. Our results are consistent with both studies, as the largest and heaviest seeds were obtained from populations G (Police) and E (Święta), formed by the highest plants, growing in large numbers and densities.

Another important factor shaping a diverse array of plant traits, including morphological features, is climate (*Nakazato, Bogonovich & Moyle, 2008*; *Colautti & Barrett, 2013*; *Van Boheemen, Atwater & Hodgins, 2019*). Temperature and precipitation gradients are the main climatic factors driving the adaptive diversification of species (*Nakazato, Bogonovich & Moyle, 2008*). As it seems, climatic conditions have had a limited effect on the investigated seed parameters till now, due to a small area of secondary distribution of *I. capensis* in Poland (*Adamowski, Myśliwy & Dajdok, 2018*; Fig. 1) and short time of residence of little over 30 years. Although this investigated plant has only a few localities inhabiting only
a relatively small area in Poland, rapid expansion across environmental gradients has been reported for several plants introduced to a new area and species can evolve quite quickly driven by environmental factors (*Dlugosh & Parker, 2008*; *Colautti & Barrett, 2013*; *Molina-Montenegro et al., 2018*; *Van Boheemen, Atwater & Hodgins, 2019*).

Phenotypic plasticity has been considered to be the primary feature enabling aliens to colonize new, environmentally diverse areas (*Richards et al., 2006*; *Molina-Montenegro, Atala & Gianoli, 2010*). However, recent research has indicated that alien plants can evolve quickly in newly occupied areas, so both phenotypic plasticity and evolution of reproductive features could be relevant factors for successful invasions (*Geng et al., 2007*; *Molina-Montenegro et al., 2018*).

An evolutionary explanation for plant invasiveness implies that seed and fruit traits are crucial for invasive plants since they are related to dispersal strategies and mechanisms to cope with environmental stress. Some research reports have indicated that native and invasive populations employ different strategies for growth and reproduction (*Chun et al., 2007*; *Molina-Montenegro, Atala & Gianoli, 2010*; *Molina-Montenegro et al., 2018*). Results by *Molina-Montenegro et al. (2018)* suggest that some seed traits of invasive plant species with rapid adaptive capacity can evolve leading to maximizing their establishment in new environments and such features can be heritable.

Due to the scarcity of data we could not point out the presence of morphological differentiation between native and invasive populations of *I. capensis*, and we have not been able to determine whether the seed traits are evolving. However, *I. capensis*, classified as an invasive species in Poland, can be suspected, while adapting and occupying new territories and competing with native species, to develop specific adaptations, contributing to its success and spread in the new environments.

## CONCLUSIONS

New data on seed morphology and seed coat sculpture of *I. capensis* is provided. The presented results are useful for the identification of this species when occurring together with other closely related species. These details on seed coat ornamentation are here described for the first time.

Further studies on the developmental variation of seed coat sculpture, especially of species closely related to *I. capensis*, may provide a better understanding of the evolutionary relationships of the different types of sculpture.

We provide new information on the plasticity of seeds of *I. capensis*. There are only few papers on the phenotypic variability of species of *Impatiens*. Data on the morphology of seeds can prove useful for determining the impact of various environmental factors on morphological traits and show whether a given feature is stable or susceptible to environmental change.

Our results suggest that certain habitat variables, especially anthropogenic disturbances and individual soil properties, contribute in shaping the morphological variation of seeds of *I. capensis*. In turn, the seed coat sculpture has turned out to be a stable feature within the secondary range of this species in Poland.

## ACKNOWLEDGEMENTS

The authors wish to express their gratitude to Theodor C.H. Cole, Dipl. rer. nat. (FU Berlin) for English language editing and valuable comments.

### Funding
The authors received no funding for this work.

### Competing Interests
The authors declare there are no competing interests.

### Author Contributions
- Agnieszka Rewicz and Monika Myśliwy conceived and designed the experiments, performed the experiments, analyzed the data, prepared figures and/or tables, authored or reviewed drafts of the paper, and approved the final draft.
- Wojciech Adamowski conceived and designed the experiments, analyzed the data, prepared figures and/or tables, authored or reviewed drafts of the paper, and approved the final draft.
- Marek Podlasiński analyzed the data, prepared figures and/or tables, authored or reviewed drafts of the paper, and approved the final draft.
- Anna Bomanowska analyzed the data, authored or reviewed drafts of the paper, and approved the final draft.

### Data Availability
  The raw measurements are provided in the Supplementary File.

### Supplemental Information
Supplemental information for this article can be found online at http://dx.doi.org/10.7717/peerj.10156#supplemental-information.

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
