# Peer review of "Seed morphology and sculpture of invasive Impatiens capensis Meerb. from different habitats"

_PeerJ, doi:10.7717/peerj.10156_

## Round 0.1 · original submission · Major Revisions

Two of the reviewers suggested improvements in your article to be considered again in PeerJ. The most important issue is that based on previous palynological analyses in Impatiens, in other members of Convolvulaceae and even in other angiosperms hypotheses should be proposed for instance on the variation of the attributes analyzed. Climate or orther ecological variables in addition to soil should be considered. Please pay attention to suggestions of these two reviewers included below and in the attached file.

·

Basic reporting

Rewicz and colleagues performed seed morphology and sculpture of Impatiens capensis Meerb. from different habitats.
The authors organized the manuscript quite well, with a good introduction, that reports information about family and the genus which Impatiens taxa belonging to, and important historical citations. Moreover, tables and figures are clear and readable.

Experimental design

Material and methods are briefly enough. Research question and aim of study well defined.

Validity of the findings

Results and discussion are adequately structured to clearly describe their findings. Conlusions are well stated.

Reviewer 2 ·

Basic reporting

Comments to author
In the present research work, the authors did work on the species Impatiens capensis seed morphology (SEM), their ultrastructure, and the relationship and variability of various environmental factors have been discussed. The paper is according to the scope of the journal, but I have some suggestions and comments on the present manuscript.
The introduction is to the point but need one paragraphs to explain the morphological and molecular difficulties of the genus Impatiens and specifically focus on the specie I. capensis.

Experimental design

In methodology some paragraphs are more looking background then the methodology. The subtitle “The Species” should be the part of introduction. Methodology is up to the point and well explained the study. This section is well explained the methodology section of the article despite with the results.

Validity of the findings

The results should be written in the past tense, change the present tense to the past tense. The results are written concise, and well explained with relation with methodology. Discussion is well explained and provide some basic information for the study of the species. I would like to suggest comparing your study with the recently published articles of other groups of plants i.e. 1) 10.1002/jemt.23393 2) 10.1002/jemt.23167
Some articles missing volume number and pages number, provide page and volume number in the revised manuscript.

Additional comments

The manuscript is well written, but i would like to suggest some major comments on the present manuscript. After that i will be agree to accept the manuscript.

Reviewer 3 ·

Basic reporting

This manuscript is clear and easy to read. The background is also enough to follow and understand the story.

Experimental design

Good experimental design.
The methods are clear.

Validity of the findings

For me, the manuscript is a bit too simple for PeerJ. It is not very interesting to me. Many publications report on seed morphology of Impatiens.

---

## Round 0.2 · Major Revisions

I appreciate that you considered previous suggestions by reviewers. However, as indicated below several reviewers suggested a number of issues that have not been yet resolved. I agree with them, for instance the Abstract does not present clearly the objective of the paper. The decision to study pollen characters to understand an invasive plant is not well presented. On the other hand no hypothesis is stated. One of the reviewers expressed concerns on selection of populations and a strong justification is needed for considering only these populations in a small area. Also figures and captions need changes. Please read and examine every issue raised by all reviewers belos.

Reviewer 2 ·

Basic reporting

I read the revised article and I am agree to accept the manuscript in current form. This manuscript is clear to understand and easy for the readers. The manuscript is have enough data to understand the scientific method of the article.

Experimental design

The experimental design is enough to understand the article.

Validity of the findings

The findings are well written and understandable.

Additional comments

I would like to accept the manuscript in current form.

Reviewer 4 ·

Basic reporting

The manuscript aims to investigate the seed morphology and sculpture of invasive species Impatiens capensis collecting from different locations. It also aims to show the effect of environmental variables on the seed features of the species. The text is well written but lacks clarity and sharpness in the discussion conclusion.

Experimental design

The methodology and statistical analyses are care, and interesting data are provided for the single species studied.

Validity of the findings

I don’t know the editor’s and journal policy, but in my opinion, this is very simple with a very narrow scope for the readers of PeerJ. As per the literature cited in the manuscript, the seed morphology of Impatiens capensis and related species is already published, and in my knowledge environmental factors including soil, characteristics have little but obvious effects on the seed morphological traits. They found no variation in ultrastructure including shape and structure of rib. The variation in seed size and mass is superficial might depend on the collection time, maturity, condition of the mother plants, and some other factors. The comparative study including some other related species would have been interesting though.

Additional comments

Specific comments are presented below, to guide the authors in the revision.
Abstract:
The abstract lacks key results and conclusions of the study.
Line 26: Our research shows also two types of cells on the ultrastructure of seeds, between the ribs and on the ribs, not yet described in the literature. I think this is a normal feature in the seed sculpture.
Line 27: The paper gives new data about the seed morphology of I. capensis growing in different habitat conditions as well as the information about the area – the circuit that has not been reported before. For me, this is a very general statement.
Same for line 29: The relationship between seed variability and various environmental factors was shown.
Introduction:
The introduction is very long. It covers the unnecessary history of the genus and species should be specific on the problems and objectives of the study. There are lots of unnecessary citations as well.
Materials and methods:
Line 149: anthropogenic? How about cultivated or human used land?
Line 152-153: the difference between almost the entire day and the entire day is confusing? What are the criteria for the condition?
Line: 167: how do you measure the thickness of the coating layer? Is it instrument-specific? I don’t think it is necessary to mention here.

Results: Tables 3 and 4 can be removed as the statistical data are presented in the text.

Discussion:
Discussion lacks clarity and sharpness. In several parts, the writing is also confusing.
Line 304-306: the seed measurement is very much comparable. The difference margin is small so it is very difficult to say the larger seed than previous reports.
Line 320-327: Writing is very confusing and vague. The discussion and comparison with I. nolitangere are superficial. The author needs to specify how the surface sculptures of both species differ.
Line 355-342: Not related to the discussion of this study. Please delete or you can adjust in the introduction part.
Line 346-367: The author discussed about the phenotypic character like height which is I think is beyond the scope of this study. Authors need to discuss how the seed morphology of I. capensis varies in different populations and what is the most influential environmental factor?
Line 368-374: I think this part is not related to the result of this study or not linked properly with the result of this study.
Conclusion: The whole part is just a general statement. I think this is not the conclusion of this study.

Reviewer 5 ·

Basic reporting

I appreciate the general structure of the paper. The subject is properly introduced and the background knowledge about the studied species is rich. However, I have several points to report.
In the abstract is explained the kind of data revealed by the study but I would expect that the authors explicitly detail what are the main results obtained in one or two sentences (regarding relation to environmental variables and seed morphology variations between populations). Moreover, I am surprised that the emphasis is put on the methodology used (SEM) but less on the questions addressed, and would suggest to better balance the abstract towards hypotheses and results.
In the manuscript, I am surprised to find very limited details about molecular differentiations between Impatiens species, although the taxonomy of this group of species is discussed.
I understand the motivations of the authors to focus on Impatiens capensis, due to its invasive nature, however I am frustrated that the study only report data about seed morphology restricted to this species without including measurement and comparison with closely related species at least, or populations from the native area to make more relevant the study.

Experimental design

The statistical analyses performed are appropriated, well conducted and explained. Figures are appropriate and well detailed (maybe check Figure 4, Ordination diagram, it seems that the asterisks are missing on the statistically significant variables, and maybe verify the caption of this figure as I do not understand what the “Anthrop” refers to, no such variable in the diagram).
My main concern is about the very limited geographical range within which the seeds have been collected. The prospected area cover only approximately 20 x 60 km, and I am concerned about the representativeness of the data (trait mean and variations) at a larger scale (country, and European range) for the species considered. To address the requirements of the journal, I would suggest including to the study seeds from more populations, either from different part of the country (to make more relevant the effect of environmental factors on seed traits) or from different ranges (to confirm your observations apply to native population in the same extend).
Indeed, as evolution of life history traits (reproductive characters) is being more and more supported as an evolutionary explanation for plant invasiveness, this aspect is important to consider, at least to discuss. Although the authors acknowledge that plant is naturalized for about 30 years in Poland, rapid evolution as been reported for several plants and can occur very quickly under the effect of climatic factors, and reproductive characters being central to invasiveness, I would expect this to be verified (Beheemen et al., 2019, doi: 10.1111/nph.15564; Molina-Montenegro et al., 2018, doi: 10.3389/fpls).
I am surprised to see that moisture content has been hand-felt assessed and wonder how precise this method is. This variable being a fundamental resource for plant development its variations may be important to consider.

Validity of the findings

The results regarding inter populations variations of seed traits with environmental factors is interesting and should be emphasized. Morphological characterization seems less relevant because no comparison with closely related species is directly made in the study. For someone intending to address taxonomical differentiation issues between Impatiens sp. it would require to perform proper comparisons (e.g. through morphometric measurements on the different species concerned).

Additional comments

In my opinion, this study brings significant and relevant results although it may miss an hindsight with the inclusion of populations from different locations.

Reviewer 6 ·

Basic reporting

Authors of the reviewed manuscript examined Impatiens capensis, an annual plant native to eastern North America which spreads across Europe and in Poland it is considered a locally invasive species. The species is an interesting and worth studying taxon, especially according to seeds which most probably play an important role in a rapid spread of that invasive species into secondary localities. Although the important differentiation in seed size and weight between examined populations was proved, the studies concerning seed biology and seedling recruitment in different environmental conditions were not undertaken. With such additional studies the work would be much more complete and would allow to evaluate the importance of seed size differentiation in spreading strategy of that invasive species. Anyway, the above mentioned problem might be an interesting aim of the future studies on I. capensis.
The actual aims of the reviewed work were as follows: the first aim was to examine seed morphology based on the standard biometric measurements; the next aim was the examination of the seed coat ultrastructure with the use of SEM; the third aim was the evaluation of the potential variability of seed morphology between individuals growing in different populations and in different habitat conditions.

Abstract:
It is not fully in agreement with the obtained results. The Authors stated in lines: 24-25 that “The current work presents a detailed description of the morphology of the seeds of I. capensis using SEM for the first time”. In my opinion such really complete description is not given in the manuscript– for the details concerning several lacks please see some of my comments given to the paragraphs Results and Discussion. Additionally, to make the Abstract more concise and clear I would suggest to unit all new and all the most important findings and underline together their taxonomic significance and/or the novelty to science.
Introduction
The Introduction is to the point, however it is sometimes too detailed in my opinion, like e.g. the information concerning the botanical description of the stem and leaves which are not the subject of the study.
Please make the text more concise and do not describe the obvious facts: e.g. lines:74-75; also devoid the repetitions: e.g. lines: 51-52 and 53; 130-132 and lines: 120-123 on the previous page; I also suggest to use proper descriptions for the 3-dimensional objects which the seeds are (instead of those suitable for the 2-dimensional ones): e.g. line 120: “seed are oval to lanceolate with …. “– instead I suggest the following description: seeds were narrowly ovate and tapering to a point at the apex; you may also use the description: seeds were laterally compressed, prolate spheroids, with four strong ribs.
There are also some inconsistences in the text e.g. in line: 93 it is written that …” One of the morphologically undescribed species is Impatiens capensis” while later in line: 125-126 the text is as follows …” Numerous studies (over several hundred – see Adamowski 2016 onward and the literature cited therein) have been devoted to the morphology, ecology, biology and genetics of this species…” – maybe the Authors meant the whole genus Impatiens, not the examined species? But it is not clean and should be corrected.
The aim of the study described in the last paragraph of the Introduction is awkward.
The English language should be improved in my opinion; it should be checked by the native speaker.
Results and Discussion:
I have listed below just few examples/observations (not all of them …) to be corrected:
- lines: 221-222: the sentences: “The seeds from the G (Police) population is characterized by the biggest seeds” and “This population characterized by the highest average values of the following traits:…” are not clear;.
- Lines: 219-220: “In the B (Lubin) population was observed the shortest and narrowest seeds (respectively:…)” again the English language correction is necessary
- The paragraph : The structure of seed surface must be rewritten according to the given below suggestions; in the present form it is not clear and the interesting results are not fully and properly described;
- The finding concerning the clear difference between seed morphology of the examined species and the closely related I. noli-tangere is important, but it would be also interesting to compare the observed type of the seed sculpture with the findings concerning other Impatiens species described by different authors (many of them were cited by the Authors, but I also recommend to see the additional following articles:

1. Three New Species of Impatiens L. from China and Vietnam: Preparation of
Flowers and Morphology of Pollen and Seeds
Author(s) :Yu-Min Shui, Steven Janssens, Su-Hua Huang, Wen-Hong Chen, and Zhi-Guo Yang Source: Systematic Botany, 36(2):428-439. 2011 and

2. 45 (5): 708–712(2007) doi:10.1360/aps06037 Acta Phytotaxonomica Sinica
Impatiens macrovexilla var. yaoshanensis S. X. Yu, Y. L.
Chen & H. N. Qin, a new variety of Balsaminaceae
from Guangxi, China.
1,2YU Sheng-Xiang 1CHEN Yi-Lin 1QIN Hai-Ning*

But to do such comparison it will be necessary to improve the present description of the seed coat micromorphology with several missing elements. Based on the observations of the structure and the microornamentation pattern of the epidermal cells of the testa, the following characters of the primary/secondary sculpture should be described or corrected: cellular/microornametation pattern should be defined; the cell outline was not clearly described; specific characters of the anticlinal walls should be described (degree of their elevation and thy type of their boundary) - by the way, from which literature comes the description anticyclic walls?; specific characters of the outer periclinal walls should be described (flat, convex, concave etc.); the secondary sculpture (if present) should also be described (e.g. cuticle striations, perhaps the presence of some protrusions etc ….).

The present descriptions of SEM images (figure captions) are not clear enough, they should be more informative, and the arrows pointing particular details should be described.

Experimental design

It is an original primary research which fits to the scope of the journal.

However the methodology needs some additional information:
- How big were the examined populations (How many individuals were growing in them approx.? Were there only few, several or more individuals, e.g. not less than 50?)
- Did you do any preparations of the seeds before examining them in SEM; were they anyhow dried? Was the seed dehydration necessary? If yes, how was it done?
- From how many individuals in each population were the seeds collected?
- line 171: instead “pictures of seed obtained using SEM’ you may write: SEM images of seeds or SEM micrographs etc. …

Validity of the findings

Concerning the variety of the examined plant material the performed studies are not too extensive, but the additional aspects concerning the relationships between the examined seed morphological characters and different environmental conditions gives the reviewed work an additional value.

The results are important and interesting. I suggest to complete and discuss the results of seed SEM analysis according to the given comments and to prepare the revised version of the manuscript. While preparing the description of the seed micromorphology and testa sculpture I would advice to follow the terminology of Barthlott W. (1981; 1984).

Additional comments

I would like to suggest Authors some minor revision of the manuscript, and encourage them to take under consideration my comments.

---

## Round 0.3 · Minor Revisions

I encourage you this last version be reviewed by a professional English editor. I read your letter and it was indicated that a native English reviewed both versions. However I found a number of problems. To mention only an example, in the Abstract it is mentioned "the presented", instead of "present". Please indicate in Acknowledements this information.

Reviewer 5 ·

Basic reporting

No more comments on this part.

Experimental design

No more comments on this part.

Validity of the findings

No more comments on this part.

Additional comments

I appreciate that the comments made on previous version of the manuscript have been taken into account. I understand that the work presented here is partial and will be included in a much larger study, coming in series of more articles. I have no more comments, and would be satisfied that the work is published as is.

---

## Round 0.4 · accepted · Accept

I appreciate that you accepted the suggestion that your paper be reviewed by a native English speaker. When the editorial office sends you a proof, my only suggestion is to explain what "Dipl. rer. nat" regarding the degree of your reviewer Theodor Cole.